# A Stringent Test of Magnetic Models of Stellar Evolution

**Guillermo Torres** [1,*] , **Gregory A. Feiden** [2] , **Andrew Vanderburg** [3] **and Jason L. Curtis** [4]

1 Center for Astrophysics Harvard & Smithsonian, 60 Garden Street, Cambridge, MA 02138, USA
2 Department of Physics & Astronomy, University of North Georgia, Dahlonega, GA 30597, USA; gregory.feiden@ung.edu
3 Department of Physics and Kavli Institute for Astrophysics and Space Research, Massachusetts Institute of Technology, Cambridge, MA 02139, USA; andrew.m.vanderburg@gmail.com
4 Department of Astronomy, Columbia University, 550 West 120th Street, New York, NY 10027, USA; jasonleecurtis@gmail.com
* Correspondence: gtorres@cfa.harvard.edu

**Abstract:** Main-sequence stars with convective envelopes often appear larger and cooler than predicted by standard models of stellar evolution for their measured masses. This is believed to be caused by stellar activity. In a recent study, accurate measurements were published for the K-type components of the 1.62-day detached eclipsing binary EPIC 219511354, showing the radii and temperatures for both stars to be affected by these discrepancies. This is a rare example of a system in which the age and chemical composition are known, by virtue of being a member of the well-studied open cluster Ruprecht 147 (age~3 Gyr, [Fe/H] = +0.10). Here, we report a detailed study of this system with nonstandard models incorporating magnetic inhibition of convection. We show that these calculations are able to reproduce the observations largely within their uncertainties, providing robust estimates of the strength of the magnetic fields on both stars: $1600 \pm 130$ G and $1830 \pm 150$ G for the primary and secondary, respectively. Empirical estimates of the magnetic field strengths based on the measured X-ray luminosity of the system are roughly consistent with these predictions, supporting this mechanism as a possible explanation for the radius and temperature discrepancies.

**Keywords:** stellar evolution; eclipsing binaries; fundamental stellar parameters; stellar activity; magnetic fields; open clusters

## 1. Introduction

The theory of stellar evolution has been one of the great success stories of 20th century Astronomy. The ability to reproduce observed properties of stars in such basic maps of Astronomy as the color-magnitude diagram or the mass–luminosity relation, and to predict unobservable properties such as their ages, have propelled our knowledge and had a significant impact on a wide array of research areas ranging from the formation of planets and stars to the evolution of entire galaxies.

While stellar astrophysics is continually advancing, so are the observational capabilities to measure fundamental properties of stars. Double-lined eclipsing binaries have traditionally been our best source of those properties, enabling masses and radii to be determined to a few percent in suitable cases, e.g., [1,2]. Such measurements are increasingly challenging stellar evolution theory. For example, it has been known for more than two decades that stars in the lower part of the main sequence (spectral types K and M) are very often larger for their mass than predicted by standard evolution models, by as much as 10%, depending on the model. This phenomenon of "radius inflation" is thought to be connected with stellar activity, which is common in these types of objects and can manifest itself in the form of strong magnetic fields (though these are hard to measure directly in binaries); star spots causing photometric variability; X-ray emission; or other spectroscopic signatures such as the lines of Hα or Ca II H and K appearing in emission. Rapid rotation is also a common occurrence. At the same time, many of the same inflated

stars display effective temperatures that are cooler than one would expect at their measured mass ("temperature suppression"), although this effect has not received as much attention because temperatures are more difficult to measure accurately.

These discrepancies are not exclusive to K and M stars. As it turns out, some active stars near the mass of the Sun, or even slightly more massive, also display enlarged radii and cooler temperatures than expected (see, e.g., [3–5]), as magnetic activity can occur more generally in any star with a convective envelope. For a detailed history and further description of the radius inflation and temperature suppression problems, we refer the reader to the reviews by [6,7]. See also the contribution by Morales and Ribas in this volume.

Over the last decade or so, efforts have been made to incorporate nonstandard physics into stellar evolution models (e.g., [8–10]) in an attempt to explain changes in the global properties of stars resulting from the actions of magnetic fields or spots associated with activity. Both of these effects tend to inhibit the convective transport of energy, and stars respond by expanding their surface area and reducing their surface temperature, in qualitative agreement with the observations. Detailed studies of individual binary systems using these nonstandard models are still relatively few in number but have shown promising results in several cases. So far, they have focused mostly on binaries with M dwarf components, for which radius inflation is more obvious.

A nagging difficulty in assessing the magnitude of the radius and temperature discrepancies for eclipsing binaries is that both of those stellar properties depend, to some extent, on the age and chemical composition of the system, which are generally not known for most of the well-measured binaries. In past studies, it has often been assumed that the metallicity is solar, for lack of a spectroscopic determination, and age choices have typically ranged between 1 and 5 Gyr, but are essentially arbitrary. The lack of better constraints on the age and metallicity makes it more difficult to make progress on understanding the impact of activity, particularly in detailed investigations with nonstandard models.

Recently, Ref. [11] reported accurate masses, radii, and effective temperatures for the K-type eclipsing binary EPIC 219511354, the subject of this paper, in which both components were shown to be significantly larger and cooler than predicted by standard models. This represents an ideal opportunity to study radius inflation and temperature suppression, because both the age and metallicity are independently known from membership of the binary in the open cluster Ruprecht 147. It presents a rare chance to test models incorporating magnetic inhibition of convection, with no free parameters other than the strength of the magnetic fields required to match the measured stellar properties. EPIC 219511354 is also more massive than the majority of objects subjected to this kind of study, which makes it particularly interesting.

The organization of this paper is as follows. Section 2 summarizes the observational properties of EPIC 219511354, as well as the constraints on the metallicity and age of the parent cluster Ruprecht 147. Magnetic models for the components are described and presented in Section 3, giving predictions for the average strength of the magnetic fields on the stars. Additional models invoking the presence of spots to explain the radius and temperature anomalies are presented here as well. We then test those predictions in Section 4, by deriving estimates for the magnetic fields based on the X-ray emission from the system and empirical power-law relations connecting various activity-related properties of stars. We conclude in Section 5 with a discussion of the results along with final remarks.

## 2. The Eclipsing Binary EPIC 219511354 in Ruprecht 147

EPIC 219511354 is one of five relatively bright, detached, double-lined eclipsing binaries identified in the old open cluster Ruprecht 147 by [12], using photometric observations gathered by NASA's K2 mission in late 2015. Four of these systems have been studied in detail in a series of papers by [11,13–15] based on the K2 light curves together with high-resolution spectroscopic observations. EPIC 219511354 is an active 1.62-day binary

with K-type components in a hierarchical triple system, with a near-circular inner orbit. We refer to the binary components as stars Aa and Ab. The unseen third star (B), possibly an M dwarf or a white dwarf, travels in an eccentric outer orbit with a period of about 220 days [11].

Evidence for activity is severalfold. The K2 light curve shows out-of-eclipse variability with a peak-to-peak amplitude of about 6%, attributed to spots on one or, more likely, both stars. The light curve residuals during primary and secondary eclipse show excess scatter that is also consistent with the presence of spots being occulted by the star in front. Both components are rapid rotators, with measured $v \sin i$ values of about 30 km s$^{-1}$ each, which imply that their rotation rates are close to those expected from spin-orbit synchronization due to tidal forces. Further evidence of activity is given by the fact that the H$\alpha$ line in the spectra is completely filled in, or slightly in emission, see [11]. Finally, EPIC 219511354 has been detected in X-rays by the Swift mission [16]. We summarize the properties of the components in Table 1, where we include the temperature difference, $\Delta T_{\text{eff}}$, which is determined directly from the light curve to higher precision than the individual temperatures.

**Table 1.** Fundamental properties of the EPIC 219511354 components [11].

| Parameter | Primary Star (Aa) | Secondary Star (Ab) |
|---|---|---|
| Mass ($M_\odot$) | $0.912 \pm 0.013$ | $0.822 \pm 0.010$ |
| Radius ($R_\odot$) | $0.920 \pm 0.016$ | $0.851 \pm 0.016$ |
| Effective temperature (K) | $5035 \pm 150$ | $4690 \pm 130$ |
| Surface gravity ($\log g$, cgs) | $4.470 \pm 0.016$ | $4.494 \pm 0.017$ |
| Bolometric luminosity ($L_\odot$) | $0.490 \pm 0.060$ | $0.316 \pm 0.038$ |
| $v \sin i$ (km s$^{-1}$) [a] | $32 \pm 3$ | $31 \pm 4$ |
| $\Delta T_{\text{eff}}$ (K) | | $345 \pm 60$ |
| Distance (pc) [b] | | $287.4 \pm 1.8$ |
| [Fe/H] (dex) [c] | | $+0.10 \pm 0.04$ |
| Age (Gyr) [d] | | $2.67 \pm 0.21$ |

[a] Measured projected rotational velocity. [b] Based on the parallax from the Gaia EDR3 catalog [17], with a zero-point adjustment following [18]. [c] Metallicity of the parent cluster Ruprecht 147 [19–21], assumed to be the same for EPIC 219511354. [d] Based on the PARSEC 1.2S models of [22].

The study of [11] compared the measured masses ($M$), radii ($R$), and effective temperatures ($T_{\text{eff}}$) of EPIC 219511354 against model isochrones from the PARSEC series [22] for the known age and metallicity of the cluster (see below); they reported that both radii are larger than predicted by 10–14% and both temperatures are cooler than expected by about 6%. The radius discrepancies are highly significant given their formal uncertainties of only 1.7% and 1.9% for the primary and secondary star, respectively. The temperature differences, on the other hand, are only significant at the $2\sigma$ level on account of the larger uncertainties, but are still suggestive given that both deviate in the same direction (opposite to the radii).

A key advantage of the EPIC 219511354 system for our purposes is its membership in Ruprecht 147, whose metallicity is well-known and is slightly supersolar, [Fe/H] $= +0.10 \pm 0.04$; [19–21]. On the assumption that the three other eclipsing binaries studied by [13–15] (EPIC 219394517, EPIC 219568666, and EPIC 219552514) all have the same composition as the parent cluster, consistent age estimates were derived independently for each of them based on the same PARSEC models mentioned above, giving a weighted average of $2.67 \pm 0.21$ Gyr. This is consistent with the result of [23] from isochrone fitting in the color-magnitude diagram of the cluster giving an age of ~3 Gyr.

Figure 1 shows the four eclipsing binaries in Ruprecht 147 that have been studied so far, plotted against the PARSEC models for this average age in both the mass–radius and mass–temperature diagrams. EPIC 219511354 clearly deviates from the best-fit isochrone, in the same direction as is often observed for other active K and M dwarfs.

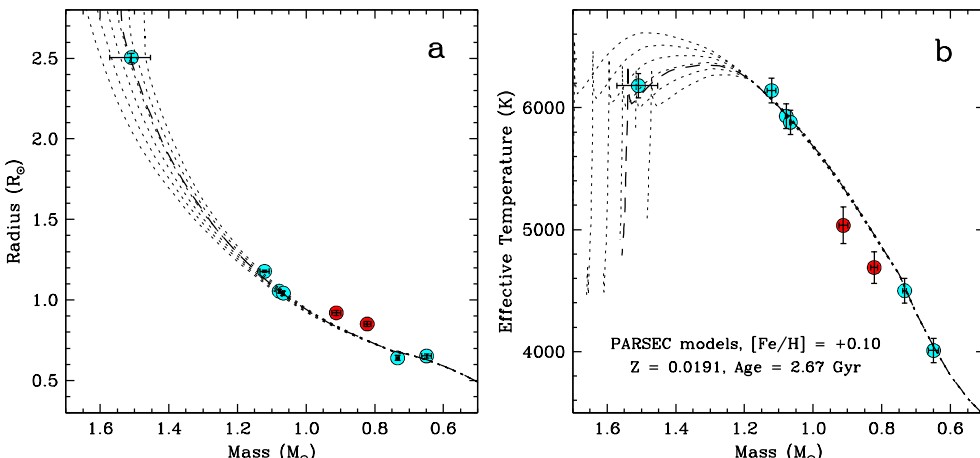

**Figure 1.** Comparison of measurements for four Ruprecht 147 eclipsing binaries against model isochrones from the PARSEC 1.2S series by [22] for the measured composition of the cluster ([Fe/H] = +0.10, $Z$ = 0.0191 in these models). EPIC 219511354 is marked in red and deviates from the best fit. (**a**) Mass vs. radius diagram. Dotted lines represent isochrones from 2 to 3 Gyr in steps of 0.2 Gyr, from the bottom up. The best-fit isochrone with an age of 2.67 Gyr according to these models is indicated with a dashed line. (**b**) Mass vs. effective temperature diagram, with the same isochrones as in the previous panel.

In contrast with the clear evidence of activity in EPIC 219511354, the other three eclipsing binaries studied previously in the cluster appear to be much less active. While they do show distortions in their light curves that are attributed to spots, the amplitude of those variations is only at the level 1% or less, far smaller than in the system discussed here. Furthermore, none show spectroscopic signs of activity such as the lines of Ca II H and K or Hα in emission, nor have they been detected as X-ray sources. EPIC 219511354 therefore stands out as the most active of the four, consistent with it being the only one that deviates significantly from the models shown in Figure 1.

## 3. Magnetic Models

With an estimated age and metallicity, EPIC 219511354 provides one of the most stringent tests of the magnetic hypothesis—that is, the hypothesis that magnetic fields or magnetic activity are responsible for observed radius and effective temperature discrepancies between stellar evolution models and measurements of fundamental parameters of low-mass stars in eclipsing binary systems. There are two primary mechanisms often discussed in this context: inhibition of convection due to the presence of strong magnetic fields [8,9], and the suppression of radiant flux due to starspots [24]. Both mechanisms produce similar effects in stellar models, but lead to different qualitative and quantitative predictions. We evaluate predictions from each mechanism.

### 3.1. Magnetic Inhibition of Convection

For testing magnetic inhibition of convection, we computed a series of stellar evolution models for EPIC 219511354 Aa and Ab from [8,25], with fixed masses ($M_{Aa}$ = 0.91 $M_\odot$, $M_{Ab}$ = 0.82 $M_\odot$) and metallicity ([Fe/H] = +0.10 dex). These models rely on the framework of the Dartmouth Stellar Evolution Program [26], with additional physics to account for perturbations from magnetic fields. For consistency, we first redetermined the age of the cluster from a best fit to the radius and effective temperature of the turnoff star EPIC 219552514 Aa [15] using standard (i.e., nonmagnetic) Dartmouth models, with some updates from the original series, as described by [25]. We obtained an age of 3 Gyr. The small difference relative to the age obtained from the PARSEC models of [22] is due to differences in the physical ingredients of the two models.

A small grid of mass tracks for each star was then created with average surface magnetic field strengths in the range 1200 G $\leq \langle Bf \rangle \leq$ 2000 G, in increments of 100 G.

Here, $B$ is the field strength and $f$ is the filling factor. Models for EPIC 219511354 Aa with $\langle Bf \rangle \geq 1700$ G did not converge as they exceed equipartition strengths near the model photosphere[1]. From these models, we constructed radius–temperature–$\langle Bf \rangle$ relationships at an age of 3 Gyr.

Standard nonmagnetic model predictions from [25] underestimate the observed radii of EPIC 219511354 Aa and Ab by 9.5% and 12%, respectively. Effective temperatures are overestimated by 4.5% and 3.8%, respectively. To establish the surface magnetic field in our models required to reproduce the observed stellar properties, we calculated a $\chi^2$ value for the primary and secondary stars independently using our derived radius–temperature–$\langle Bf \rangle$ relationships at 3 Gyr. We then found the $\langle Bf \rangle$ values that minimized the total $\chi^2 = \chi^2_{\text{radius}} + \chi^2_{\text{temp}}$ value ($\chi^2_{\text{min}}$) for each stellar component. Approximate uncertainties in the model $\langle Bf \rangle$ values were determined by satisfying $\chi^2(\langle Bf \rangle) = \chi^2_{\text{min}} + 1$.

Models with magnetic inhibition of convection predict $\langle Bf \rangle_{\text{Aa}} = 1500 \pm 150$ G and $\langle Bf \rangle_{\text{Ab}} = 1850 \pm 150$ G. Results are shown in Figure 2a,b for EPIC 219511354 Aa and Ab, respectively. The ratio of surface magnetic field strengths is $\langle Bf \rangle_{\text{Aa}} / \langle Bf \rangle_{\text{Ab}} = 0.81 \pm 0.10$, indicating that the secondary has the stronger field. We find that models with magnetic inhibition of convection are able to reproduce the observed properties of EPIC 219511354 Aa and Ab within $1\sigma$ of their formal errors (see Table 1), with the exception of the secondary's effective temperature. Magnetic models able to reproduce the secondary star's radius predict effective temperatures that are systematically too cool compared with the measured effective temperature by 3.4% ($1.2\sigma$).

We may improve our fit to the secondary star's effective temperature by calculating a joint $\chi^2$ value for the two components. The total $\chi^2$ value is the sum of the $\chi^2$ values for the individual radii, the primary star's effective temperature, and the temperature difference $\Delta T_{\text{eff}}$ from Table 1. This yields a slightly stronger surface magnetic field strength for the primary star, $\langle Bf \rangle_{\text{Aa}} = 1600 \pm 130$ G, and a marginally weaker strength for the secondary star, $\langle Bf \rangle_{\text{Ab}} = 1830 \pm 150$ G. These results are fully consistent with those from fitting the individual components.

Our results vary little for adopted ages between about 2.5 and 3.5 Gyr. Model properties do not change appreciably over this 1-Gyr time span, and the magnetic model evolution parallels standard model evolution. Furthermore, the results are also quite insensitive to the adopted metallicity, provided it is within the measurement errors. This indicates that our $\langle Bf \rangle$ values are robust. However, if the age and metallicity of EPIC 219511354 were instead left completely free, the results could be significantly different, emphasizing the importance of having those constraints in this case.

### 3.2. Starspots

Stellar models that incorporate changes to stellar properties due to the presence of starspots were tested using SPOTS models [24,27]. The SPOTS model grid has a fixed, solar metallicity and a fixed temperature ratio between spots and the ambient photosphere of 0.80. However, it does allow for variations in the starspot surface coverage between spotless and 85% [27]. We estimated the predicted surface coverage of starspots for each component of EPIC 219511354 at an age of 3.0 Gyr.

We find that SPOTS models require a surface coverage $f_{\text{spot}} = 0.59 \pm 0.04$ and $f_{\text{spot}} \approx 0.91 \pm 0.03$ to fit the observed properties for the primary and secondary, respectively. We were able to interpolate within the SPOTS grid to estimate the surface coverage for the primary star but we were forced to extrapolate from the grid to estimate the surface coverage for the secondary. It is evident that the expected surface coverage for EPIC 219511354 Ab is $f_{\text{spot}} > 0.85$, although the precise value may differ from our extrapolated estimate.

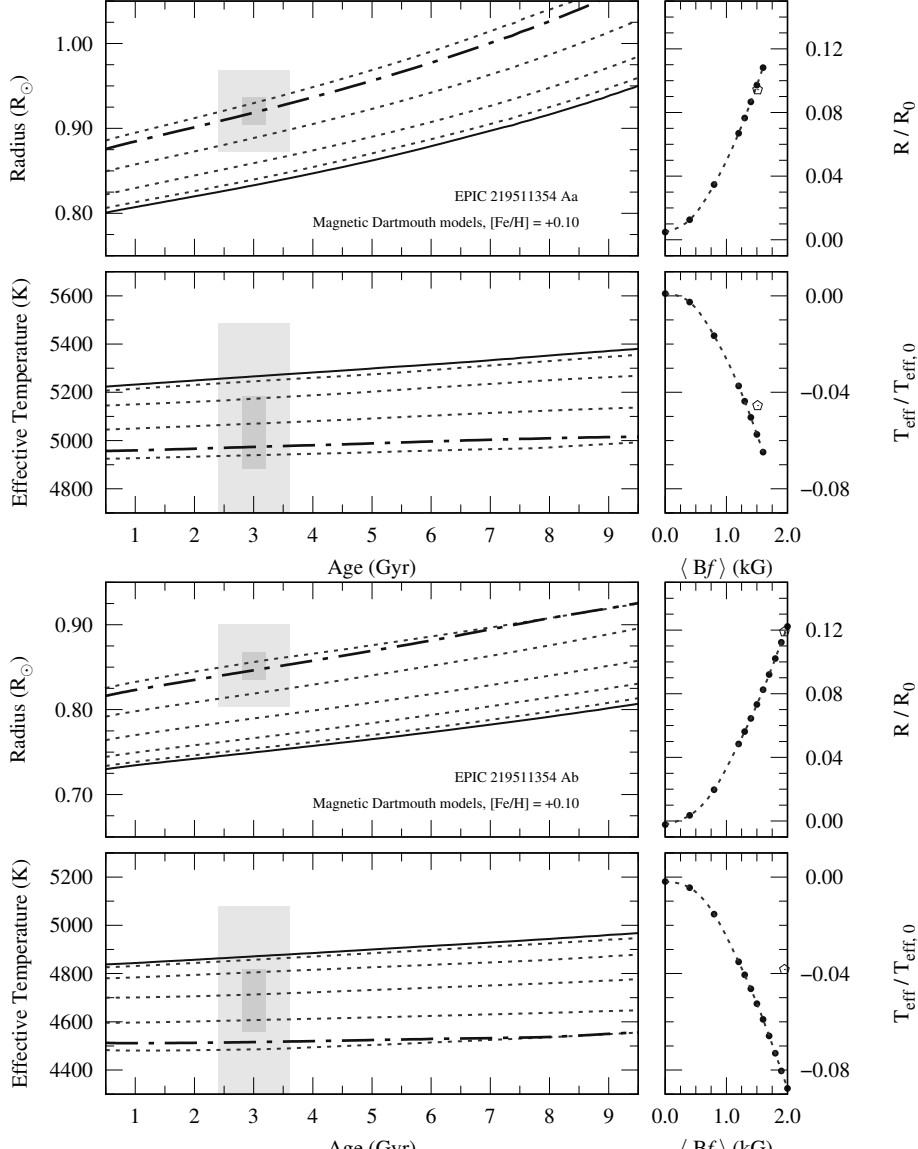

**Figure 2.** Estimating the average surface magnetic field strength for EPIC 219511354 Aa (**top panels**) and Ab (**bottom panels**). For each star, we plot the model radius and model $T_{\rm eff}$ as a function of time for standard stellar evolution models from [25] (black lines), magnetic stellar evolution models with $\langle Bf \rangle$ in increments of 400 G (dotted lines), and best-fit magnetic stellar evolution model (dash-dotted lines). Gray shaded regions represent the $1\sigma$ and $3\sigma$ uncertainties surrounding the measured values. Side panels illustrate how the model radius and effective temperature change compared with the standard model as a function of $\langle Bf \rangle$. Open symbols show the observed radius and effective temperature deviations with the best-fit $\langle Bf \rangle$ values.

From the starspot surface coverage values, we can estimate the average surface magnetic field strength for the individual components. If one assumes that starspots are in pressure equipartition with the surrounding photospheric gas, the expected temperature ratio between starspots and photospheric gas should be approximately 80% for stars similar to EPIC 219511354 Aa and Ab. Therefore, we can estimate that the stars are covered by a fraction $f_{\rm spot}$ of equipartition-strength magnetic fields. Equipartition values are $B_{\rm eq,\, Aa} \approx 1700$ G and $B_{\rm eq,\, Ab} \approx 2000$ G, based on estimates of the gas pressure at $\tau_{\rm ross} = 1$ from MARCS model atmospheres [28]; thus, SPOTS models predict $\langle Bf \rangle_{\rm Ab} \approx 1000$ G and $\langle Bf \rangle_{\rm Ab} \approx 1800$ G, for a ratio $\langle Bf \rangle_{\rm Aa} / \langle Bf \rangle_{\rm Ab} \approx 0.56$. This ratio is formally lower than that we obtained from the models with magnetic inhibition of convection.

Limitations imposed by the SPOTS grid mean that we are unable to provide an accurate constraint on the required starspot properties. In particular, the adopted temperature ratio between starspots and the ambient photosphere is fixed at 80% in these models. A ratio of 80% is reasonable for pressure equipartition, but this ratio could be as low as 70–75% based on energy equipartition[2]. A lower ratio implies cooler spots, reducing the required surface coverage to about 40% and 70% for Aa and Ab, respectively. As a result, the surface coverages quoted above should be interpreted as upper limits on the actual values.

## 4. Empirical Estimates of the Magnetic Field Strength

The detection of EPIC 219511354 as an X-ray source by the Swift mission gives us a means to quantify, in an approximate way, the strength of the magnetic field on each component of the binary, if we assume that the contribution of the tertiary is negligible. This is an important check on the predictions from the previous section based on magnetic models. The 0.3–10 keV X-ray flux reported by [16] is $F_X = 3.70 \pm 0.59 \times 10^{-13}$ erg s$^{-1}$ cm$^{-2}$. When combined with the distance listed in Table 1, it leads to a total X-ray luminosity of $L_X = 3.71 \pm 0.57 \times 10^{30}$ erg s$^{-1}$. To make use of this observational constraint, we proceeded as follows.

We first appealed to a power-law relation reported by [29] (Figure 1 in that work) between the average surface magnetic field strength, $\langle Bf \rangle$, and the Rossby number, $\mathrm{Ro} \equiv P_{\mathrm{rot}}/\tau_c$. Here, $P_{\mathrm{rot}}$ is the rotation period and $\tau_c$ is the convective turnover time, which [29] obtained from a relation by [30] as a function of $T_{\mathrm{eff}}$. The [29] relation, $\langle Bf \rangle \propto \mathrm{Ro}^{-1.2}$, has considerable scatter, so we initially applied it only in a differential sense, making use of the slope to calculate the ratio of the magnetic field strengths. As the components are rotating very close to their synchronous velocities, we have assumed that $P_{\mathrm{rot}} = P_{\mathrm{orb}}$, leading to $\langle Bf \rangle_{\mathrm{Aa}}/\langle Bf \rangle_{\mathrm{Ab}} = (\tau_{c,\mathrm{Ab}}/\tau_{c,\mathrm{Aa}})^{-1.2} = 0.828 \pm 0.035$. This suggests the secondary component has the stronger average magnetic field strength, and the ratio agrees well with the prediction from models with magnetic inhibition of convection described above.

Next, we used a result from a study of magnetic field observations of the Sun and active stars by [31], who showed that there is a fairly tight power-law relation $L_X \propto \Phi^p$ between the X-ray luminosity and the unsigned surface magnetic flux, $\Phi = 4\pi R^2 \langle Bf \rangle$, which holds over many orders of magnitude. [25] provided an update of this relation specifically for dwarf stars, giving $p \approx 2.2$. With the ratio of the radii known (Table 1), we computed the X-ray luminosity ratio as $L_{X,\mathrm{Aa}}/L_{X,\mathrm{Ab}} = 0.93 \pm 0.14$ and, with the total luminosity derived earlier, we inferred the individual values $\log L_{X,\mathrm{Aa}} = 30.22 \pm 0.12$ and $\log L_{X,\mathrm{Ab}} = 30.25 \pm 0.08$, with $L_X$ in the usual units of erg s$^{-1}$. The relation by [31], with updated parameters from [25], finally leads to individual estimates of the magnetic field strengths of $\langle Bf \rangle_{\mathrm{Aa}} = 1100 \pm 150$ G and $\langle Bf \rangle_{\mathrm{Ab}} = 1300 \pm 160$ G.

More direct estimates of $\langle Bf \rangle$ may also be obtained via the full [29] relation and, although they are formally consistent with those above, the results, $\langle Bf \rangle_{\mathrm{Aa}} = 2000 \pm 840$ G and $\langle Bf \rangle_{\mathrm{Ab}} = 2400 \pm 1100$ G, are considerably more imprecise because of the scatter of the [29] relation.

These approximate field strengths are in line with those of other active K dwarfs (see, e.g., [32]) and are also consistent with an empirical relation between $\langle Bf \rangle$ and the rotational velocity shown in Figure 21 of that paper, for the measured $v \sin i$ values of EPIC 219511354.

Both sets of estimates are roughly consistent with the values required by the magnetic models from the previous section to match the radius and temperature measurements for the binary components of EPIC 219511354. In particular, our most precise empirical estimates are within about 2 or 2.5$\sigma$ of the predicted values from theory.

Finally, with bolometric luminosities computed from the radii and temperatures listed in Table 1, we may also calculate the X-ray to bolometric luminosity ratios, $\log(L_X/L_{\mathrm{bol}})_{\mathrm{Aa}} = -3.36 \pm 0.12$ and $\log(L_X/L_{\mathrm{bol}})_{\mathrm{Ab}} = -3.33 \pm 0.08$, indicating that the activity in both components is at the saturation level (see, e.g., [33]).

## 5. Discussion and Final Remarks

Radius inflation and temperature suppression in convective stars are now well-established phenomena. Mass–radius diagrams—and less often, mass–temperature diagrams—shown in many papers on this subject are being populated by growing numbers of K and M dwarfs in eclipsing binaries, boosted in recent years by new systems discovered in photometric searches for transiting planets, including those from space. While the radius and temperature deviations are seen clearly from the ensemble of measurements, relatively few of the systems with well-measured properties, i.e., mass and radius uncertainties below 3% (see, e.g., [2]), have been subjected to detailed studies with models incorporating nonstandard physics, in an attempt to explain those discrepancies. This would seem to be the next logical step to understand the nature of the deviations.

Among the main-sequence binaries examined with the methodology and magnetic models employed in this paper, five examples with components above the fully convective boundary (corresponding to masses of about 0.30–0.35 $M_\odot$) have been studied so far. They are UV Psc, YY Gem, CU Cnc, EF Aqr, and V530 Ori [8,25,34]. Their properties can all be adequately explained with surface magnetic field strengths $\langle Bf \rangle$ ranging from 1.3 to 4.6 kG, depending on the system, which are consistent for the most part with empirical estimates based on X-ray emission in the cases where such observations exist. In some systems, these predictions depend to some degree on the poorly known age and metallicity of the binaries, among other factors. Our study of EPIC 219511354 adds another example to this short list, with the important distinction that both age and metallicity are well-known, which removes free parameters and makes the test of these models more demanding. We find that the predicted strengths of the magnetic fields needed to explain the observed radius inflation and temperature suppression are within about 40% of our most precise empirical estimates (a 2–2.5$\sigma$ difference, provided both the theoretical and observational errors are realistic).

However, the ability of magnetic models of the same lineage as those applied here to explain the properties of fully convective M dwarfs, including CM Dra and Kepler-16 [35], has been less satisfactory. The predicted field strengths required to reproduce the observations tend to be larger than is thought to be plausible for such objects, particularly for stars with the largest deviations compared with standard models.

Several other binary systems, both above and below the fully convective boundary, have been studied with a different prescription for incorporating the effects of magnetic fields, e.g., [9,36,37]. Not all of these systems are eclipsing (i.e., with direct radius measurements) and, in at least one case, the precision of the mass and radius measurements is rather poor, but a general conclusion one draws is that the predicted strengths for the magnetic fields tend to be lower than with the models used in this paper. For example, for YY Gem, a system with partially convective components, the predicted $\langle Bf \rangle$ values from [36] are a factor of 3–4 lower than those measured directly by [38] using Zeeman–Doppler imaging techniques, whereas those by [25] are some 20–40% larger. For an in-depth description of the differences between these models, see [25,39].

The results for EPIC 219511354 in this paper seem encouraging and suggest that modeling efforts are on the right track. On the observational side, dozens of mass and radius measurements are now available for K and M dwarf eclipsing binaries, but the quality of the data is by no means uniform. As has been pointed out before (e.g., [6]) systematic errors in the measurements are likely an important, if not dominant, contributor to the scatter in the mass–radius diagram for cool main-sequence stars. In most cases, this is caused by the ubiquitous presence of spots on these active stars, which can bias the results, particularly for the radii (see, e.g., [40,41]). Painful evidence of these biases can be seen by comparing results for eclipsing systems for which two or more independent studies have been made. T-Cyg1-12664 [42–44], NSVS 07394765 [45,46], and PTFEB132.707+19.810 [47,48] are three recent examples where significant differences in the reported component masses and/or radii are seen among authors, who sometimes even made use of some of the same photometric or

spectroscopic observations. Part of these disagreements could also be due to the application of different methodologies for the analysis.

The overall scatter seen in recently published mass–radius diagrams for K and M dwarfs leads us to believe that the utility of such diagrams for understanding the impact of activity on the global properties of these kinds of stars has probably reached its limit. We no longer advance our knowledge much by simply adding more objects to the diagram, some of which may be of questionable quality. It seems more likely that more progress can be made by focusing on individual systems with the best-measured properties, ideally with known ages and metallicities, for which estimates can be made of the strength of the activity either by inference from the X-ray flux, as in EPIC 219511354, or preferably from direct measurements via Doppler tomography or other methods. This has been a main motivation for this paper. Quantifying the activity in one of those ways is essential for testing the predictions of magnetic models, or of models with spots. Eclipsing binaries in open clusters, though relatively uncommon, are a promising source of constraints on such models, as the age and metallicity may be known from the parent population, as in the case of EPIC 219511354.

**Author Contributions:** Conceptualization, G.T. and G.A.F.; formal analysis, G.T. and G.A.F.; investigation, G.T., G.A.F., A.V. and J.L.C.; writing—original draft preparation, G.T. and G.A.F.; writing—review and editing, G.T., G.A.F., A.V. and J.L.C. All authors have read and agreed to the published version of the manuscript.

**Funding:** This research was funded by NASA/ADAP grant number 80NSSC18K0413.

**Institutional Review Board Statement:** Not applicable.

**Informed Consent Statement:** Not applicable.

**Data Availability Statement:** Not applicable.

**Acknowledgments:** We thank the anonymous referees for helpful comments.

**Conflicts of Interest:** The authors declare no conflict of interest. The funders had no role in the design of the study; in the collection, analysis, or interpretation of data; in the writing of the manuscript, or in the decision to publish the results.

## Notes

[1]　We are referring specifically to pressure equipartition, which is defined as the condition that the isotropic magnetic pressure equals the surrounding gas pressure, $P_{\mathrm{mag}} = B^2/8\pi = P_{\mathrm{gas}}$.

[2]　Here, the magnetic field's energy density is equal to the internal energy density of the gas. This permits magnetic field strengths about ~20% stronger than in pressure equipartition.

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
