# Peer review of "A Stringent Test of Magnetic Models of Stellar Evolution"

_galaxies, doi:10.3390/galaxies10010003_

Round 1

Reviewer 1 Report

In the series of the papers Torres et al. analysed eclipsing binaries in the open cluster Ruprecht 147. In the Paer IV (ApJ, ) Torres et al. revealed an active eclipsing binary with the K-type components showing anomalies in the radii and effective temperatures in sense disclosed for late type stars with inflated radii, and cooler temperatures than predicted by standard evolutionary models. In the present manuscript the authors examined two possible explanations (1) magnetic inhibition of convection, and (2) starspots. The authors found the preeceding explanation more plausible with calculations better reproducing the observations, including measured X-ray luminosity. The paper is well, and clearly written, and address hot topic in the observational stellar astrophysics. Before final recommendation for its publication, I would like the authors respond on the following comments, and suggestions:

1) The effective temperature and radii are the parameters showing deviations to standard stellar models. As the authors are pretty much aware inconsistency in the effective temperatures are propagating also to the radii through the light ratio. I did check in the Paper IV of their series how the effective temperatures for the stars in the binary in quest are detrmined. I am sure that determination of the effective temperatures would be more solid if the authors used separation of the observed spectra. Since they measured the radial velocities for both components it is straightforward to make tomographic separation. This process has many advantageous including enhancement in S/N of the separated spectra.

2) The authors are speaking on precise determination of the temperature difference between the components. If I understand correctly the temperatures are determined in the spectral range of Mg I triplet at 5180 A, while bandpass transmission of Kepler photometry has maximum in red and IR spectral range. Thus I expect some systematics in the difference of effective temperatures.

3) In Table 1 fundamental properties of the components in binary system EPIC 219511354 are listed. It might be more appropriate to indicate that metallicity quoted is determined for cluster, not directly for the stars in binary system, just as the age, and distance are from other sources.

4) In line 123 the authors refer to age determined fro isochrone fitting. It would be more useful for potential reader to give number from ref. [9], to avoid unnecessary looking for it in quoted reference.

5) Some of the authors conclusions rely on the assumption that stars in binary system(s) bear the same metallicity as stars in the cluster. Very probably this assumption holds, but in a view it is rather central assumption in their analysis of eclipsing binary systems in the open cluster Ruprecht 147 it might be appropriate to put it on a firm ground. Along my comments ad (1) the authors would profit from separation of the observed spectra, and abundance analysis of the separated spectra. Atmospheric and detail abundance analysis of separated spectra of binary stars have advantage over spectroscopic analysis of single stars since the surface gravities are determined much more precisely, which further constrain the analysis. From recent spectroscopic studies of Ruprecht 147 (Curtis et al.2013, 2018; Bragaglia et al., 2018) I see spread in [Fe/H] from about -0.1 to 0.2. Detail abundance study using the stars in binary systems, with more precise determined atmospheric parameters might narrow this spread.

6) What might be role of non-standrad model atmospheres with incorporated magnetic inhibition of conection on determination of the effective temperatures?

Reviewer 2 Report

This is an excellent paper that certainly merits to be published by Galaxies. The authors address an important issue of low mass stars in binary systems that has been standing for long time, i.e. the inflated stellar radii observed compared to model predictions for the same masses. The use of a system with excellent absolute dimensions belonging to a cluster constraints the age and metallicity of the models and allows a better comparison with predictions based on magnetic inhibition of convection or starspots.

As a minor comment, the level of activity in the systems of Ruprecht 147 included in Figure 1 (in blue) should be mentioned to clarify the special case of EPIC 219511354. 
